# Genetic Diversity of Creole Sheep Managed by Indigenous Communities of the Central Region of Veracruz, Mexico

**DOI:** 10.3390/ani12040456

**Published:** 2022-02-13

**Authors:** Ruth Guadalupe Castillo-Rodríguez, Obdulia Lourdes Segura-León, Martha Hernández-Rodríguez, Ricardo Serna-Lagunes, Josafhat Salinas-Ruiz, Juan Salazar-Ortiz

**Affiliations:** 1Maestría en Innovación Agroalimentaria Sustentable, Colegio de Posgraduados, Campus Córdoba, Veracruz 94953, Mexico; castillo.ruth@colpos.mx (R.G.C.-R.); salinas@colpos.mx (J.S.-R.); 2Colegio de Posgraduados, Campus Montecillo, Texcoco 56230, Mexico; sleon@colpos.mx (O.L.S.-L.); hernandez.martha@colpos.mx (M.H.-R.); 3Facultad de Ciencias Biológicas y Agropecuarias, Región Orizaba-Córdoba, Universidad Veracruzana, Veracruz 94945, Mexico; rserna@uv.mx

**Keywords:** microsatellites, heterozygosity, allelic diversity, endogamy, consanguinity

## Abstract

**Simple Summary:**

The genetic diversity in three populations of creole sheep managed by indigenous communities of the central region of Veracruz, Mexico, is reported from blood samples of 90 sheep, taken from the herds of indigenous families in the municipalities of Tehuipango, Astacinga, and Tlaquilpa, Veracruz, Mexico. The genomic DNA of the sheep was evaluated using four microsatellites amplified by PCR and visualized on polyacrylamide gels. The four microsatellites were polymorphic, observed heterozygosity was lower than the expected level, and the indices of endogamy indicated a slight diminution of homozygotes and the variation was hosted at the individual level. The evaluated sheep present a genetic diversity that is conserved across endogamic crosses, for which reason the design of a plan of protection and use of these sheep populations would permit their sustainable management.

**Abstract:**

In the indigenous communities of central Veracruz, herds of creole sheep have been established and managed through traditional practices of crossing, but their genetic characteristics have never been examined in order to evaluate their state of endogamy, and to help the management programs to protect this genetic resource. The objective of the present study was to characterize the genetic diversity of three populations of creole sheep managed by indigenous communities in the central region of Veracruz, Mexico. Indigenous family producers of creole sheep were located and blood samples taken from 90 individual sheep from the municipalities of Tehuipango, Astacinga and Tlaquilpa, Veracruz. In the laboratory, the genomic DNA was extracted and genetic diversity characterized using four microsatellites (ILSTS11, ILSTS5, SRCRSP9 and OarFCB128) amplified by PCR and visualized on polyacrylamide gels. The four microsatellites were highly informative (PIC = 85%) and presented values of 0.6 to 0.81 of heterozygosity, with an average number of 16 alleles. According to the Hardy–Weinberg equilibrium model, three of the loci were not significant (*p* < 0.05), presumably this means that they do not deviate significantly from H–W predictions and there was slight genetic differentiation (*F_ST_* = 0.025), along with a slight decrease in homozygotes (*F_IS_* = −0.021). According to the analysis of variance, 99% of the total variation was hosted at the individual level. It is concluded that the three creole sheep populations still present genetic diversity at the four loci and non-random pairings have occurred.

## 1. Introduction

Livestock production was introduced to Mexico during the colonial period and has become a socioeconomically productive activity [1]. Ovine livestock are managed in a wide range of productive systems for meat, dairy products, skins, and other derivatives [2]. The eastern state of Veracruz is the third largest producer of ovine livestock, and is organized into three production districts, north, center, and south, with subsistence production (42%), commercial subsistence (21%), and commercial (37%) systems [3]. The most common breeds of sheep in these districts are crosses derived from Pelibuey, Black Belly, and Dorper sheep, as well as creole breeds [4]. 

The creole sheep breeds are those populations that have adapted to the rural and cultural environments in which the herds are managed but still present certain traits of the original breeds introduced to Mexico during the colonial period [5]. Among the individual creole sheep, it is possible to observe a wide range of morphotypes arising from the innate genetic variability of the sheep and the diversity of climatic and geographic conditions of each rural region in which they are managed [6]. However, few studies have explored the extent of the genetic diversity of the creole sheep morphotypes that are observed in the rural regions of Mexico [7].

Study and diagnosis of the genetic diversity of creole sheep are necessary in order to determine the status of the genetic variability of the populations [8], take measures to avoid genetic depression of the sheep populations that make them vulnerable to biotic and abiotic factors [9], improve the response to changes in their environment [10], and develop plans and strategies for genetic conservation and improvement [11]. There is also interest in the study of genetic diversity as a basis for the development and use of ovine production systems that could strengthen the food sovereignty of indigenous communities [12]. In this sense, the objective of this study was to characterize the genetic diversity of three populations of creole sheep from the localities of Tehuipango, Astacinga, and Tlaquilpa, in central Veracruz, Mexico, using molecular marker type microsatellites.

## 2. Materials and Methods

### 2.1. Characteristics of Populations of Sheep Evaluated and Blood Sample Collection 

The populations of sheep (*n* = 90 sheep) under study were found in the municipalities of Tehuipango (population 1, *n* = 33 sheep, 11 males and 22 females), Astacinga (population 2, *n* = 21 sheep, 4 males and 17 females) and Tlaquilpa (population 3, *n* = 36 sheep, 11 males and 25 females), located in the central region of Veracruz, Mexico (Figure 1). The creole sheep specimens presented a background and a mixture of phenotypic characters associated with the breeds Churra, Laza, Castellana, Canaria, and Manchega (Figure 2). The indigenous families that managed the sheep herds were located and permission sought to take blood samples from their animals. The blood sampling process followed the normal ethics of animal management [13] and 3 to 4 mL of whole blood were extracted by puncture of the jugular vein using the B-D Vacutainer Plus equipment with EDTA (Becton-Dickinson & Company; Nueva Jersey, EU) [14]. The animals were treated in accordance with the ethical standards and technical specifications of the Official Mexican Standard and technical specifications for the production and humane treatment in the mobilization of animals (NOM-051-ZOO-1995) and use of laboratory animals (NOM-062-ZOO-1999).

### 2.2. DNA Extraction and PCR

DNA was extracted from each blood sample using the Wizard^®^ Genomic DNA Purification Kit (Promega, Madison, WI, EU). Genotyping of each individual was conducted with the following four microsatellite type molecular markers: OarFCB128, SRCRSP9, ILSTS5, and ILSTS11 (Table 1). These microsatellites were chosen from the panel of markers recommended by the FAO (2011), based on their polymorphic information content (PIC) value. Amplification of the microsatellites was conducted by PCR in a final reaction volume of 20 µL comprising the following components: 40 ng of DNA, buffer 1×, 0.1 µM of each primer, 1.5 mM of MgCl2, 0.16 mM of each dNTP, 0.6 U of Taq polymerase (Promega, Madison, WI, EU) and PCR-grade water to reach the final reaction volume. The program of amplification for ILSTS11 and ILSTS5 consisted of an initial denaturation step at 94 °C for 2 min, followed by 30 cycles of 94 °C for 60 s, 53 °C for 60 s, 72 °C for 60 s, and a final extension step of 72 °C for 5 min. For SRCRSP9 and OarFCB128, the program followed the same steps, with the exception of the annealing temperatures, which were 50 °C and 56 °C, respectively. Amplification of each microsatellite was verified in 1.5% agarose gel.

### 2.3. Electrophoresis of the Microsatellites

To determine the size in bp of the product of each amplification, electrophoresis was conducted in non-denaturing polyacrylamide gels and using 3 µl of each reaction per well in gels prepared with 19:1 acrylamide:bisacrylamide [20] in the MGV-216-33 vertical electrophoresis system (CBS Scientific^®^, California, USA), with the following parameters: migration at 250 V for 1.5 h, running buffer TBE 1X (0.09 M Tris-borate, 2 mM EDTA pH 8.0), and 25 ng of 20 bp molecular weight marker (Sigma-Aldrich) placed every 10 samples. Each fragment was detected using silver staining [21]. Each gel was documented in the format *.TIFF in a MiniBis Pro16 mm transilluminator (Bio Imaging Systems^®^, Jerusalem, Israel). The weight of each band in bp was determined with the software GelAnalyzer version 19.1 in order to construct the matrix of molecular data.

### 2.4. Analysis of Genetic Data

The Hardy–Weinberg Equilibrium (HWE) of each marker was evaluated using a Chi-squared test, and the markers were also examined in order to qualify their polymorphism and obtain the parameters of population diversity [22]. These parameters were: allelic frequencies at 5% reliability, number of alleles per locus (Na), effective number of alleles (Ne), expected (He) and observed (Ho) heterozygosity and indices of fixation F (*F_IS_*, *F_IT_* and *F_ST_*) as well as analysis of molecular variance (AMOVA), which were processed with the software packages GenAlEx V.6.503 [22] and PowerMaker V.3.25 [23]. The software DarWin V. 6.0 [24] was used to construct a tree of genetic relationships based on the mean Euclidean distances between pairs of sheep populations.

## 3. Results

This study found significant deviations from the HWE in some of the loci studied, and suggested non-random pairings in the populations of creole sheep. ILSTS11 locus was the only one in the HWE loci/marker or similar between only and in the three populations, while the rest of the microsatellite loci were found to be in HWE in only one of the three populations: locus ILSTS5 was found in equilibrium for population 1 (Astacinga) and the loci OarFCB128 and SRCRSP9 were found in equilibrium in population 2 (Tehuipango) (Table 2). The four microsatellites (OarFCB128, SRCRSP9, ILSTS5, and ILSTS11) were polymorphic since more than one allele is present among all of the loci analyzed within a population. The polymorphism values of the four genotyped microsatellites were thus found to be 0.83, 0.88, 0.84, and 0.85 for ILSTS11, ILSTS5, SRCRSP9, and OarFCB128, respectively, and selection of these loci for genotyping the three populations of sheep was therefore highly informative.

The allelic frequencies per microsatellite (Table 3) as the locus ILSTS11 presented 15 different alleles in the three populations, ILSTS5 locus presented 20 different alleles, SRCRSP9 locus presented 13, and OarFCB128 locus presented 12 different alleles. Minimum and maximum size of the alleles ranged from 260 to 294 bp for ILSTS11, 186 to 216 bp for ILSTS5, 108 to 130 bp for OarFCB128, and 90 to 200 bp for SRCRSP9. 

Table 4 presents the statistics of genetic diversity of the loci in the three populations of creole sheep. This table lists the markers, frequency of the most common allele within the populations, number of genotypes, sample size, number of alleles observed, genetic diversity, heterozygosity, and PIC value. 

The degree of genetic differentiation (*F_ST_*) was 0.025 (ILSTS11), 0.011(ILSTS5), 0.029 (SRCRSP9), and 0.034 (OarFCB128) revealing that the sheep of the high mountains of central Veracruz present what is considered low genetic differentiation, suggesting minimal changes in the genetic structure of the population (Table 5). On the other hand, the coefficient of endogamy (*F_IS_*) for each locus was negative: −0.234 (ILSTS11), −0.150 (ILSTS5), −0.248 (SRCRSP9), and −0.221 (OarFCB128), all one can conclude is there is an excess of heterozygotes, indicating a greater incidence of non-random pairings among related individuals. 

In this study, the phylogram showed that the Nei genetic distances ranged from 0.062 to 0.089, where two groups with similar genetic characteristics (27%) were identified. The first of these were grouped as population 1 (Astacinga) and population 2 (Tehuipango), indicating that these populations presented a high degree of genetic similarity, while the second group was represented by population 3 (Tlaquilpa), which was genetically differentiated from populations 1 and 2 (Figure 3).

Finally, the AMOVA showed that the greatest genetic variation was provided by the individuals of the population (99%), and that there was almost null variation (1%) among individuals of the different sheep populations evaluated (Figure 4).

## 4. Discussion

The HWE showed that the frequencies of the microsatellites evaluated in the three populations of sheep have not remained stable among generations, since these sheep originate from a population with a low number of founding individuals, or that experienced little random breeding. The lack of equilibrium detected in the loci could have been motivated, in part, by the need of indigenous producers for crossing with rams that present phenotypic characteristics of commercial value [25] or by the family selection typical among producers in the region. From this perspective of allelic frequencies, it is clear that genetic inheritance and variation in the creole sheep is beginning to be affected by at least one of the evolutionary forces, presenting a situation that endangers the genetic variability that has been produced through adaptation to the conditions of the high mountains of the central region of Veracruz and that, in terms of survival, disease resistance, regional identity, etc., has been fundamental for these sheep and their producers.

### 4.1. Polymorphism of the Loci

The four microsatellite loci evaluated in the creole sheep presented more than one allele [26]. The microsatellites evaluated in this study to genotype the three populations of sheep were highly informative, since the PIC values were > 0.50 [27]. Study of the loci represented a tool that, in addition to distinguishing individuals at the genetic level, was useful to determine the small population differentiation that was detected. With this, the informative value of the selected loci could be applied to other creole sheep populations that develop in indigenous production systems, in order to establish a pedigree that helps with decision making regarding genetic improvement and the conservation of animal germplasm.

### 4.2. Allelic Diversity and Number of Alleles 

Studies of genetic diversity have reported values of allelic diversity and allele numbers similar to those found in this study. In a population of Nigerian sheep, alleles of 96 to 130 bp were recorded at the loci OarFCB128 [16]. In a study of genetic diversity and paternity in a population of goats, reported alleles of 119 to 200 bp for the microsatellite SRCRSP9 [17], on other study reported allele sizes of 262 to 292 bp for ILSTS11 and 107 to 133 bp for SRCRSP9 in a study of genetic characterization of sheep in Colombia [28]. In a study of zoometric and genetic characterization of sheep in Brazil using the microsatellites ILSTS11, ILSTS5, and SRCRSP9, reported sizes that ranged from 266 to 288 bp, 181 to 201 bp, and 99 to 135 bp, respectively [29]. In a study of genetic characterization of Colombian creole sheep, also used the microsatellites ILSTS11, ILSTS5, and SRCRSP9, obtaining allele size ranges of 262 to 292 bp, 176 to 214 bp, and 107 to 133 bp, respectively [30]. A study of genetic diversity and relationships of indigenous breeds of creole sheep in Pakistan using microsatellite loci, with ILSTS11 and OarFCB128 among the markers used, and reported allele size ranges of 300 to 382 bp and 106 to 136 bp, respectively [31].

The numbers of alleles observed per locus in this study exceeded those reported in a study of characterization of Colombian creole sheep, reported 11 different alleles for ILSTS11, 12 alleles for ILSTS5, and 12 alleles for OarFCB128 [28]. The values found in the present study also exceed those reported by [31], who identified just two alleles for the locus OarFCB128 and four for ILSTS11 in two different populations of creole sheep. The presence of 14 different alleles for ILSTS11, 16 for ILSTS5, and 10 for SRCRSP9 [28], and another study that reported lower allele numbers than those reported in this study, analyzed genetic polymorphism and bottlenecks in sheep of Balkhi [32]. In this case, the study reported the presence of only two different alleles for the markers ILSTS11 and ILSTS5, three alleles for the marker SRCRSP9, and four for the marker OarFCB128. In an evaluation of diversity and genetic differentiation in breeds of Portuguese thick-wooled sheep using microsatellite markers, included the markers ILSTS5 and ILSTS11, for which eight alleles were found [33]. For this reason, the number of alleles found in this study suggests that the creole sheep of Astacinga, Tehuipango, and Tlaquilpa present high allelic diversity, perhaps due to the constant flow of individuals among the herds. 

Within genetic improvement programs, knowledge of the genetic diversity of creole sheep is the basis for reproductive and productive success in populations for sustainable management [34]. In the case of heterozygosity, which is considered one of the parameters of diversity that best represent variability within a population [35], the values were 0.60 (ILSTS11), 0.81(ILSTS5), 0.76 (SRCRSP9), and 0.77 (OarFCB128), with an average of 0.73. The heterozygosity of the individuals is associated with greater performance and adaptability on the part of the individuals of the populations [36]. In this context, when individuals present low heterozygosity, they have lower aptitudes and probabilities of survival in a given medium and of creating progeny, for which reason the value of this parameter is used as an indicator of endogamy within and among populations [37]. In this sense, the degree of heterozygosity in the sheep evaluated in this study suggests that, for the purpose of use, population heterozygosity could move in a manner directed towards the biocultural conditions of higher-yield management systems.

In this study, the *F_ST_* index revealed that the creole sheep present low genetic differentiation, suggesting a genetic structure that is becoming unstable at population level [4]. This means that the evaluated animals are still genetically similar within each population, but there are already signs of apparent introgression. This result was supported by the low and negative value of the coefficient of endogamy (*F_IS_*) for the four microsatellites evaluated, indicating a decrease in homozygous individuals. In a study of genetic characterization of sheep in Colombia, reported *F_IS_* values higher than 0, which was attributed to the fact that the breeds could be inbred or subjected to human selection through management in order to acquire certain phenotypic characteristics [28]. The sheep populations studied revealed deficiency in heterozygosity and genetic variation individual, resulting from the reproductive tendency of these populations [32].

The results also identified two groups: the first of which included population 1 (Astacinga) and population 2 (Tehuipango), while the second consisted only of population 3 (Tlaquilpa). The group that included populations 1 and 2 showed that these populations have a high degree of genetic similarity, which can be attributed to the fact that the physical proximity of their regions has facilitated the occurrence of crossing between these populations. Finally, it is possible that the allelic variability of the creole sheep evaluated has been selected as a function of the indigenous group or family that manages the herd, which has produced the small genetic differentiation. The creole sheep populations should still maintain sections of genome that comprise a large part of their identity; however, given the limited number of loci examined, this situation remains to be demonstrated.

## 5. Conclusions

Genetic diversity, based on the genotyping of four microsatellite markers in three populations of creole sheep managed by indigenous communities in the central region of Veracruz, showed a large catalog of alleles. The results contribute to the genetic knowledge of these sheep required in order to control situations through reproductive management. For the purposes of survival, it is essential to protect and maintain these populations. Furthermore, for the purposes of use it is required to fix phenotypic characteristics such as wool production in the populations of Astacinga and Tehuipango; rams and ewes of these populations that express these desired phenotypic characteristics should be crossed. Moreover, if it is desired to increase the height of the cross, rams from Tlaquilpa can transfer this characteristic over a few generations, given the degree of endogamy presented by the populations.

## Figures and Tables

**Figure 1 animals-12-00456-f001:**
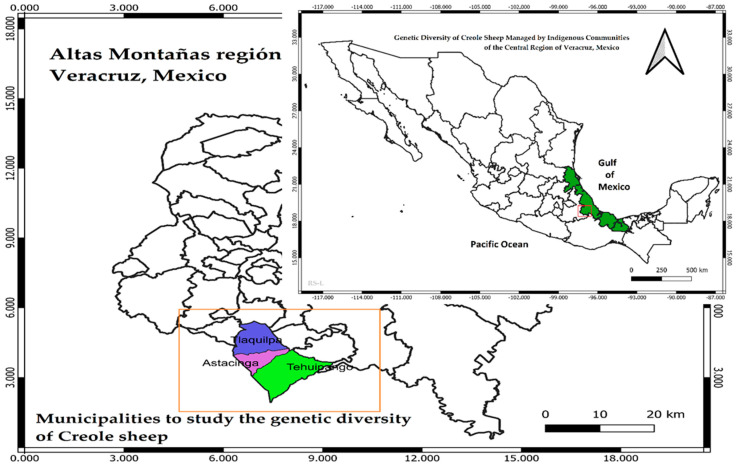
Localities of the sheep populations under study in the central region of Veracruz, Mexico. These populations are referred to as: population 1 (Astacinga), population 2 (Tehuipango), and population 3 (Tlaquilpa). Source: prepared by the coauthors with data collected from the field. Copyright permission.

**Figure 2 animals-12-00456-f002:**
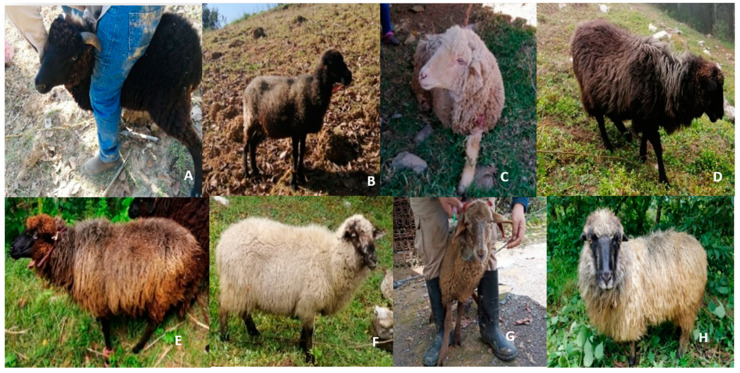
Creole sheep sampled in the central region of Veracruz, Mexico: (**A**) creole black sheep, (**B**) creole black/white/brown sheep, (**C**) creole white sheep, (**D**) creole grey sheep, (**E**) creole brown/blackface sheep, (**F**) creole black/white/brown sheep morphotype two, (**G**) creole brown sheep, and (**H**) creole white/grey/black sheep.

**Figure 3 animals-12-00456-f003:**
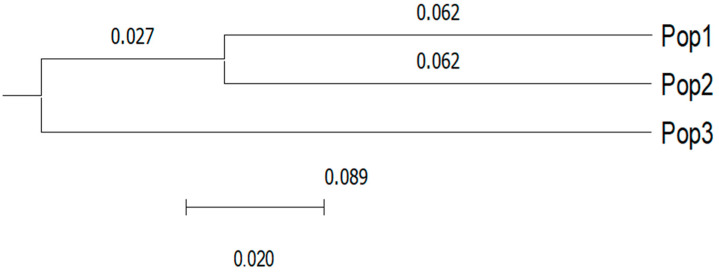
Phylogenetic relationships of three populations of creole sheep in the central region of Veracruz, Mexico.

**Figure 4 animals-12-00456-f004:**
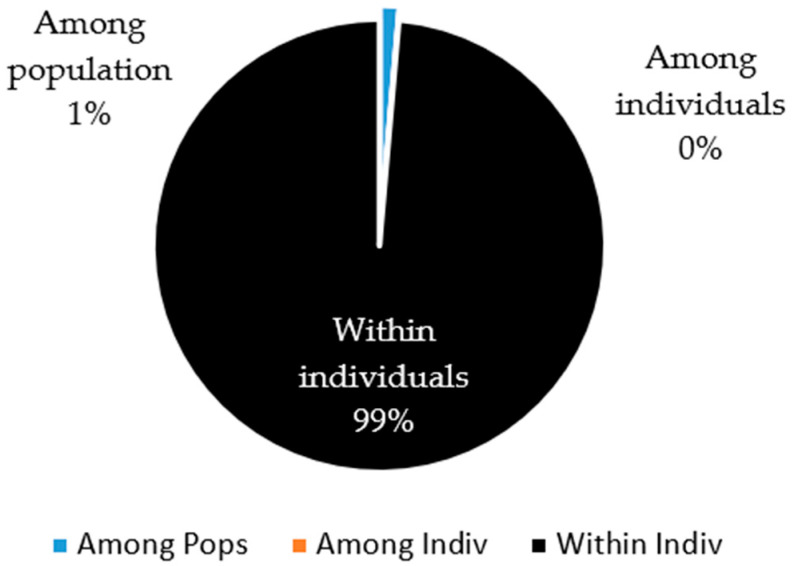
Partition of the genetic diversity among individuals, within individuals, and among populations of creole sheep using microsatellites.

**Table 1 animals-12-00456-t001:** Primers used to describe the genetic diversity of creole sheep from central Veracruz, Mexico [15].

Locus	Sequence 5′–3′	Size (bp)	Source
OarFCB128-F	ATTAAAGCATCTTCTCTTTATTTCCTCGC	96–136	[16]
OarFCB128-R	CAGCTGAGCAACTAAGACATACATGCG		
SRCRSP9-F	AGAGGATCTGGAAATGGAATC	119–143	[17]
SRCRSP9-R	GCACTCTTTTCAGCCCTAATG		
ILSTS5-F	GGAAGCAATGAAATCTATAGCC	174–221	[18]
ILSTS5-R	TGTTCTGTGAGTTTGTAAGC		
ILSTS11-F	GCTTGCTACATGGAAAGTGC	256–294	[19]
ILSTS11-R	CTAAAATGCAGAGCCCTACC		

**Table 2 animals-12-00456-t002:** Test of Hardy–Weinberg equilibrium for four microsatellites examined in three populations of creole sheep in central Veracruz, Mexico.

Population	Locus	df	X^2^	*p*-Value	Significance
Pop1	ILSTS11	36	23.833	0.940	ns
Pop1	ILSTS5	66	69.686	0.355	ns
Pop1	SRCRSP9	10	26.071	0.004	**
Pop1	OarFCB128	28	52.308	0.004	**
Pop2	ILSTS11	36	41.382	0.247	ns
Pop2	ILSTS5	66	148.901	0.000	***
Pop2	SRCRSP9	45	56.012	0.126	ns
Pop2	OarFCB128	36	49.965	0.061	ns
Pop3	ILSTS11	36	41.348	0.248	ns
Pop3	ILSTS5	45	68.033	0.015	*
Pop3	SRCRSP9	55	88.053	0.003	**
Pop3	OarFCB128	28	80.500	0.000	***

df = degree of freedom, X^2^ = Chi squared, ns = non-significant, * *p* < 0.05, ** *p* < 0.01, and *** *p* < 0.001. * Markers not found in HWE (*p* < 0.05).

**Table 3 animals-12-00456-t003:** Allelic frequencies of the microsatellite markers studied in three populations of creole sheep in Veracruz, Mexico.

Marker	Allele	Pop 1	Pop 2	Pop 3
ILSTS11	260	0.000	0.030	0.000
ILSTS11	268	0.048	0.015	0.028
ILSTS11	270	0.048	0.000	0.028
ILSTS11	274	0.024	0.000	0.000
ILSTS11	276	0.048	0.000	0.056
ILSTS11	278	0.190	0.182	0.167
ILSTS11	279	0.000	0.015	0.000
ILSTS11	280	0.262	0.318	0.222
ILSTS11	282	0.000	0.091	0.125
ILSTS11	284	0.214	0.136	0.167
ILSTS11	286	0.000	0.015	0.056
ILSTS11	288	0.024	0.000	0.000
ILSTS11	289	0.000	0.030	0.000
ILSTS11	290	0.024	0.030	0.014
ILSTS11	294	0.119	0.136	0.083
ILSTS5	186	0.048	0.000	0.000
ILSTS5	188	0.000	0.015	0.000
ILSTS5	190	0.143	0.076	0.111
ILSTS5	192	0.000	0.015	0.000
ILSTS5	193	0.000	0.015	0.000
ILSTS5	194	0.024	0.076	0.069
ILSTS5	195	0.000	0.000	0.014
ILSTS5	196	0.143	0.152	0.083
ILSTS5	198	0.119	0.061	0.042
ILSTS5	200	0.167	0.212	0.208
ILSTS5	214	0.024	0.000	0.000
ILSTS5	202	0.000	0.015	0.000
ILSTS5	204	0.024	0.076	0.083
ILSTS5	206	0.024	0.030	0.069
ILSTS5	208	0.024	0.061	0.014
ILSTS5	210	0.119	0.121	0.056
ILSTS5	212	0.048	0.045	0.000

**Table 4 animals-12-00456-t004:** Statistics of genetic diversity of three populations of creole sheep in central Veracruz, Mexico (*n* = 90).

Marker	Frequency of the Most Common Allele	No. of Genotypes	No. of Alleles	He	Ho	PIC
ILSTS11	0.27	34	16	0.85	0.60	0.83
ILSTS5	0.20	33	20	0.89	0.81	0.88
SRCRSP9	0.21	29	14	0.85	0.76	0.84
OarFCB128	0.17	21	15	0.86	0.77	0.85
Average	0.22	29.25	16.25	0.86	0.73	0.85

He= expected heterozygosity, Ho = observed heterozygosity, and PIC = polymorphic information content.

**Table 5 animals-12-00456-t005:** Indices of genetic differentiation and of endogamy (*F*) for the microsatellites studied in populations of creole sheep in central Veracruz, Mexico.

Locus	*F_IS_*	*F_IT_*	*F_ST_*
ILSTS11	−0.234	−0.203	0.025
ILSTS5	−-0.150	−0.137	0.011
SRCRSP9	−0.248	−0.212	0.029
OarFCB128	−0.221	−0.18	0.034
Mean	−0.213	−0.183	0.025

## Data Availability

Not applicable.

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
