# Peer review of "Genetic Diversity of Creole Sheep Managed by Indigenous Communities of the Central Region of Veracruz, Mexico"

_animals, 2022, doi:10.3390/ani12040456_

Round 1
Reviewer 1 Report
The authors describe a small study investigating creole sheep diversity in a region of Mexico using only four microsatellites and three populations. The authors report diversity measures and a low differentiation between all three populations. While relatively small, the paper is technically accurate and well documented for replication and validation. Below are minor edits to the manuscript to improve clarity.
L76: Materials & Methods
Please list out the number of animals that were sampled from each population. There is no mention of numbers of animals used from each group. Were the animals all the same sex?
L85: add "whole" before blood and remove "cardiac" since you are not puncturing the heart
L111: I would re-title the table "Primers used..." since it lists the primers and not the microsatellite repeats themselves
L139: "was the only in HWE" please add loci/marker or similar between "only" and "in"
L143-145: Codominance is an inheritance pattern and as such can not be determined from loci information alone. I would remove that section of the sentence and leave it as just described as polymorphic.
L164: The formatting of the table appears slightly off i.e. the bolding of marker ILSTS11, please review/revise if appropriate
L172: From the negative FIS value, all one can conclude is there is an excess of heterozygotes. This could be the result of other factors and not solely non-random pairings.
Author Response
"Please see the attachment."

Reviewer 2 Report
Castillo-Rodrigues and colleagues have analysed the variation in four microsatellites in Mexican creole sheep managed by indigenous communities. They observe generally low heterozygosity, suggesting patterns of non-random mating, consistent with cultural herd management practices.
It would be good to tidy up some of the language for the sake of clarity. eg. ln 36 "three of the loci were not significant" - at what? Presumably this means that they do not deviate significantly from H-W predictions, but it would be good to say so.
The final clause in the sentence that spans lns 27 and 28 appears to belong to the following sentence, and makes no sense where it is.
ln 59 - versatility? I think perhaps this should be variability.
I'm not sure whether this is a problem at the author-level or the editorial-level, but the text and the figures bleed into each other, particularly in the results, making it very hard to read. Perhaps some of the tables and figures can be moved to the supplementary material?
The conclusions strike me as rather odd - as though the authors ran out of things to write and so just wrote down some random thoughts. Having the two final sentences of the manuscript start with "For example..." is just strange. This section needs a re-think.
Author Response
"Please see the attachment."
